# Automatic and accurate ligand structure determination guided by cryo-electron microscopy maps

Andrew Muenks [1,2], Samantha Zepeda [1], Guangfeng Zhou [1,2], David Veesler [1,3] & Frank DiMaio[1,2] ✉

Advances in cryo-electron microscopy (cryoEM) and deep-learning guided protein structure prediction have expedited structural studies of protein complexes. However, methods for accurately determining ligand conformations are lacking. In this manuscript, we develop EMERALD, a tool for automatically determining ligand structures guided by medium-resolution cryoEM density. We show this method is robust at predicting ligands along with surrounding side chains in maps as low as 4.5 Å local resolution. Combining this with a measure of placement confidence and running on all protein/ligand structures in the EMDB, we show that 57% of ligands replicate the deposited model, 16% confidently find alternate conformations, 22% have ambiguous density where multiple conformations might be present, and 5% are incorrectly placed. For five cases where our approach finds an alternate conformation with high confidence, high-resolution crystal structures validate our placement. EMERALD and the resulting analysis should prove critical in using cryoEM to solve protein-ligand complexes.

Recent advancements in both microscope hardware and computational processing have led to cryo-electron microscopy (cryoEM) emerging as a mainstream method for biomolecular structure determination. While in ideal cases cryoEM data approaches atomic resolutions[1–3], most structures determined by cryoEM are in the 3–5 Å resolution range. At these resolutions, model building is time-consuming, error prone, and often ambiguous. To assist this process, methods have been developed to automatically build de novo polypeptide chains into EM data[4–7], and with the advent of AlphaFold 2, high-quality starting models can oftentimes be obtained from sequence information alone[8,9]. While these methods help build protein models into cryoEM density, tools for automatic fitting of small molecule ligands into cryoEM data are limited. Given the widespread adoption of cryoEM in academia and in industry to support translational studies of drug targets, the ability to accurately model ligand-bound structures is paramount.

There are numerous automated tools from X-ray crystallography for modeling small molecule ligands[10–13]. However, their methodology is unproven for use in interpreting all but the highest-resolution cryoEM maps. Traditional ligand fitting methods rely on shape and topological features of density maps to match[10] or build[11–13] the ligand into density. But as resolution decreases below 3 Å, the topological features these methods rely on become less defined, and their accuracy in modeling ligands within 1 Å RMSD of a reference ligand falls below 20%[13,14]. While these software packages have been updated to consider cryoEM data, the updates focus on protein modeling without reported updates to small molecule modeling[15,16] or focus on small molecule refinement instead of automatic model building[17].

Along with map features, chemical force fields have provided an energetic approach to accurately fit ligands into their respective density. Two approaches—GemSpot[18] and MDFF[19]—utilize the ligand-docking software GLIDE to model ligands into cryoEM data. How-

[1]Department of Biochemistry, University of Washington, Seattle, WA 98195, USA. [2]Institute for Protein Design, University of Washington, Seattle, WA 98195, USA. [3]Howard Hughes Medical Institute, University of Washington, Seattle, WA 98195, USA. ✉e-mail: dimaio@uw.edu

ever, both require user input in either selecting models during the protocol or choosing a starting configuration, limiting the automation and applicability of these approaches. The protein modeling software Rosetta recently incorporated a small molecule force field, RosettaGenFF, which accurately models the energetics of arbitrary biomolecules in a manner balanced against Rosetta's protein force field[20]. Combining this energy model with a genetic-algorithm (GA) optimization method allowing for full receptor side chain flexibility, GALigandDock, yielded superior performance in ligand docking accuracy compared to other state-of-the-art methods.

Here, we leverage the docking power of RosettaGenFF and GA optimization to overcome the challenges of modeling small molecules at near-atomic resolution. We integrate cryoEM density data with the physically realistic force field of RosettaGenFF to create RosettaEMERALD (EM Maps ERoded for Automatic Ligand Docking) for robust ligand modeling into cryoEM maps with no user input during the protocol. We evaluate the performance of EMERALD on all non-ion-mediated ligand-bound protein structures deposited in the EMDB[21] and compare our results to their respective deposited structures and high-resolution crystal structures when available.

## Results

An overview of EMERALD is illustrated in Fig. 1. GALigandDock places ligands in a protein pocket by iteratively refining a pool of 100 conformations, selecting the best 100 models at each generation using predicted energy. To enable this method to use cryoEM density, two changes were integral: density-guided initial ligand placement and the use of density in model selection at each round. Our initial placement (fully described in Methods) first models density as a pseudo-atomic skeleton (Fig. 1b). When generating the initial population of ligands, ligands are placed at the center of the skeleton and restrained to points in the skeleton. At each iteration, the population of ligand conformers along with their surrounding flexible side chains are further optimized against the sum of a weighted density correlation and the RosettaGenFF energy (Fig. 1c) and finally refined in Rosetta to minimize the energy of the models (Fig. 1d). The full protocol generates a structure in 30–120 min, depending on the size of the ligand and the cryoEM map.

To test EMERALD, we ran our docking protocol on all ligands with 25 or fewer rotatable torsion angles present in deposited cryoEM structures determined at a minimum of 6 Å nominal resolution. This yielded 1053 ligands to be placed. For each model, we ran three independent trajectories, and we analyzed the resulting models using three different criteria: (a) agreement of the deposited model to the lowest energy predicted structure; (b) density fit and number of protein/ligand hydrogen bonds; and (c) convergence of the three trajectories. This last criterion is used to evaluate the confidence in a predicted model.

The results of these docking trajectories are summarized in Fig. 2. In 57% of the cases, our density-guided docking produced a top model within 1 Å RMSD (considering all non-hydrogen atoms in the ligand) of the deposited model after energy minimization (match, Fig. 2a). While an RMSD cutoff of 2 Å has traditionally been used for docking success, the lack of confidence in the low-resolution reference models and inability of RMSD to consider receptor contacts led us to divide results further by density correlation and hydrogen bond contacts. There were 401 cases (38%) where EMERALD produced a model with an RMSD value >1 Å, and the model was similar or better than the deposited model in both metrics (non-match, similar or better quality). The smallest group belonged to 48 cases where the deposited model was not recapitulated, but the EMERALD model had a worse density fit or fewer hydrogen bonds than the deposited model (non-match, worse quality, 5%). Modeling accuracy decreases as ligand flexibility increases and as the local resolution of the map surrounding the ligand worsens (Fig. 2b, c). Also, we found that incorporating EM data in GALigandDock is necessary for recapitulating deposited ligand structures with high success rates (Supplementary Fig. 1).

Because of the low resolution of the density maps, it is difficult to interpret the quality of docked poses from density fit and receptor interactions alone. To instill more confidence in docking results, we analyzed the convergence among the top-ranked ligand poses across three replicates (Fig. 2d–f). Of the cases within 1 Å RMSD, 2 or more of the trajectories converge for 81% of cases, further strengthening the quality of the matched cases (Fig. 2d). Moreover, only 23% of the worse-quality cases converge on the same ligand model (Fig. 2e). Given how well trajectory convergence agrees with these categories, it can serve as a proxy for confidence when our docked model differs from the reference model in ambiguous cases. 42% of the ambiguous cases have similar top models across our trajectories (Fig. 2f), giving us confidence in an alternative model to the deposited structure for those entries.

Our dataset includes 15 of the 20 cases benchmarked for the GemSpot pipeline[18], with five cases filtered out of the dataset for being peptides or having inter-residue bonds like ion coordination. For 13 of the 15 ligands, EMERALD produced a ligand within 1 Å of the deposited structure, with nine of those placements assessed as confident. For the

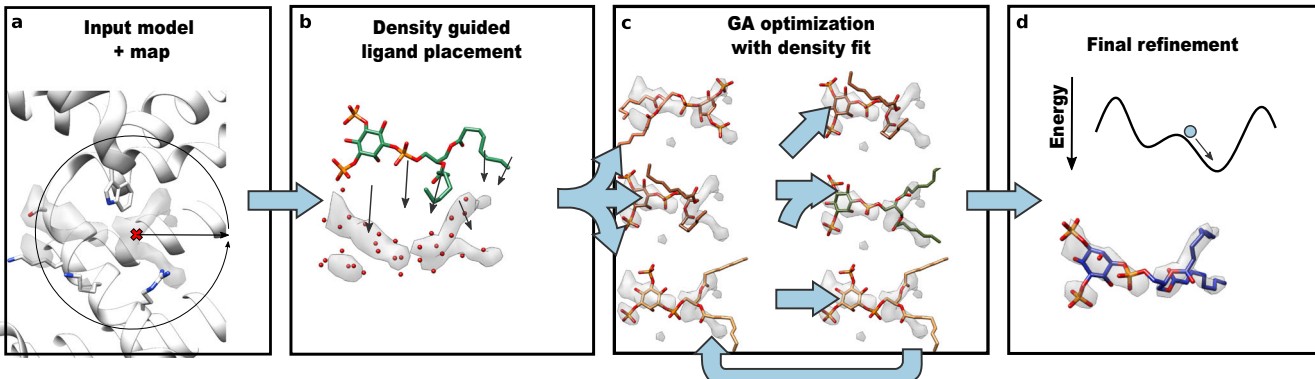

**Fig. 1 | Overview of EMERALD docking protocol. a** The cryoEM map, coordinates of the receptor, and the location of the binding site (red cross) are provided as inputs. The binding pocket is calculated depending on the radius of the ligand (circle) to determine boundaries and side chains to consider when modeling. **b** All unmodeled density in the pocket is converted to a pseudo-atomic skeleton (independent of ligand identity), which is used to generate an initial set of ligand conformers. **c** Using a genetic algorithm, the pool of ligand conformers is optimized against Rosetta energy and density fit. The population of ligand conformers evolve over 10 generations with low energy conformations surviving and combining attributes with each other. **d** The 20 poses with lowest energy are refined in Rosetta.

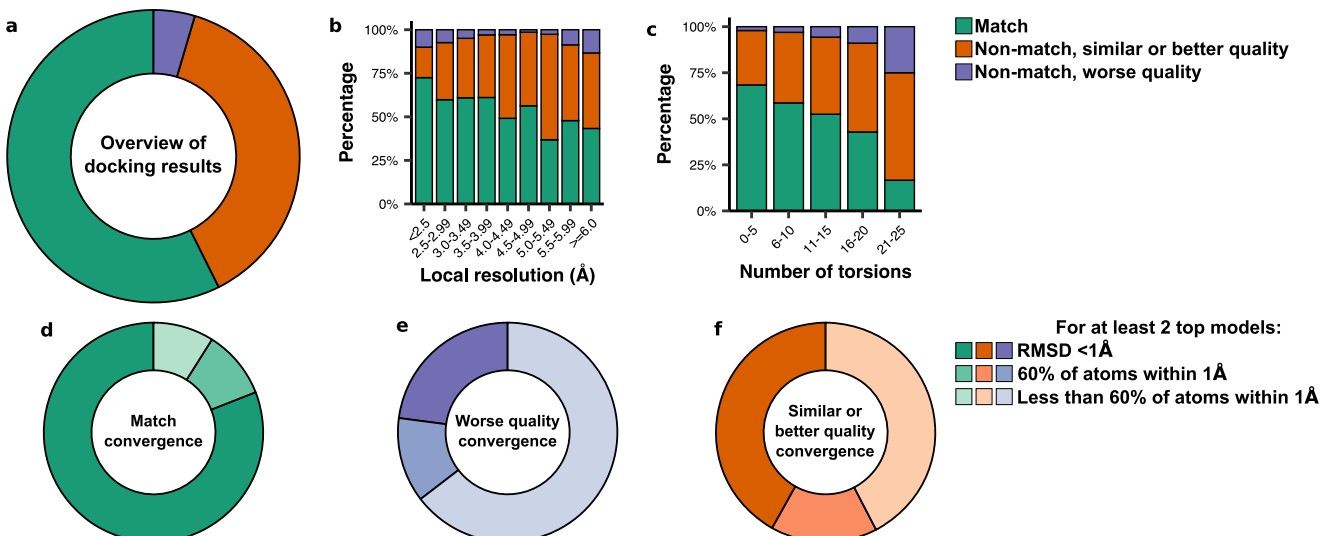

**Fig. 2 | Benchmarking EMERALD against the EMDB. a** A comparison of EMERALD models to the deposited structures for 1053 EMDB-deposited complexes. In total, 57% of EMERALD-docked models were placed within 1 Å RMSD of the deposited ligand (match, green); 38% were more than 1 Å RMSD of the deposited ligand but had similar or better density correlations and numbers of hydrogen bonds (similar or better quality, orange), and 5% were more than 1 Å RMSD from the deposited ligand and had worse density correlations or number of hydrogen bonds (worse quality, blue). **b**, **c** Bins of binding pocket local resolution (**b**) and number of torsion angles in the small molecule (**c**) shown as percentage by docking result. **d**–**f** The convergence of the best ranking models across multiple runs for matches (**d**), worse quality (**e**), and similar or better quality (**f**) cases. The darkest shade had multiple runs converge with all atoms within 1 Å of each other, the middle shade had multiple runs converge with at least 60% of atoms within 1 Å, and the lightest shade had divergent top-scoring models.

other two cases, our models disagreed with the deposited model; GemSpot also found solutions different from the deposited model in these two cases.

## Crystal models confirm alternate conformations for EM data

To cross-validate our results—particularly in cases where we found a different solution than the deposited model—we looked for all models with a corresponding high-resolution crystal structure (see Methods). We identified 100 cases where EMERALD converged on a ligand placement and a corresponding high-resolution crystal structure was available. The converged docked model was within 1 Å RMSD of the ligand modeled in the crystal structure for 67% of cases, while 58% of the deposited EM models were within this distance. Considering cases where the model predicted from EMERALD and the reference EM model differ, there were six cases where the EMERALD model was within 1 Å RMSD to the crystal structure while the EM model was not, three cases where the EM model was within 1 Å of the crystal structure but the EMERALD model was not, and eight cases where both models differed from the crystal structure by more than 1 Å. In addition, in five of the six cases where our model predicts the crystal structure, our ligand model improves density correlation by at least 0.03, compared to the deposited cryoEM model.

We show docked models supported by crystal structures in Fig. 3 to highlight the quality of our protocol. These examples include: (a) the hippocampal AMPA receptor with the antagonist MPQX[22], where our model makes additional hydrogen bond and π-stacking interactions with the ligand, matching the crystal structure[23] (Fig. 3a); (b) NBQX in an AMPA receptor[24], where the ligand is flipped, better matching the density, and making bidentate interactions with a nearby arginine residue (Fig. 3b); (c) DNMDP bound to the SLFN12-PDE3A complex[25], where small changes better match the crystal structure (Fig. 3c); (d) an ADP molecule in ClpB disaggregase[26] (Fig. 3d), where the phosphate groups recapitulate the crystal structure; and (e) a glutamate ligand in the AMPA glutamate receptor[24], which was missing an oxygen atom in the deposited structure; when the full glutamate molecule is docked, the carboxylates are placed in a configuration matching the crystal structure[27] (Fig. 3e).

There were three cases where our docking protocol found a ligand different than the crystal structure, while the EM model matched the crystal structure closely. All three cases were different maps of the same system, a folate molecule bound to MERS-CoV[28,29]. In all 3, the EMERALD model and the crystal structure only differ in the placement of a flexible arm with high B-factors in the crystallographic data (Supplementary Fig. 2)[30]. These results lend more support for EMERALD convergence as a confidence metric, which we used to further find instances of alternate ligand conformations.

## Docked poses reveal plausible alternate conformations

Even without crystal structures for reference, trajectory convergence and improved ligand density fit provide confidence in other docked poses. In the case of an antimicrobial bound multiple transferable resistance (Mtr) pump[31], our protocol converges on an ampicillin molecule that is flipped so that its phenyl group is now in a pocket of unassigned density (Fig. 4a, b). While the deposited model places the phenyl group sandwiched between two phenylalanine residues (Fig. 4a), our docked model packs the group near a cluster of hydrophobic residues known to interact with other antibiotics[31] (Fig. 4b). In addition, nearby arginine, serine, and threonine residues have been suggested to generally coordinate ligands binding to the pump[31]; our model has the carboxyl group positioned to make interactions with these residues directly or possibly through bridging water molecules. While it is likely that an antibiotic would bind non-specifically to this site, EMERALD ranks our presented orientation the highest across all three trajectories, and there is a large predicted energy gap (about 10 kcal/mol) between the converged conformation and the best-scoring conformation with the phenyl group outside this hydrophobic pocket, suggesting that this pose is strongly favored by EMERALD.

Another instance of improving density fit and receptor interactions is a lipid phosphatidylinositol 4,5-bisphosphate ($PIP_2$) bound to transient receptor potential melastatin member 8 (TRPM8)[32]. The EMERALD docked model correlates with the map 10% better than the deposited model, placing all the phosphate groups into density and placing the likely disordered glycerol backbone and beginning of the lipid tails in weaker density (Fig. 4c, d). Moreover, the 4,5 phosphate

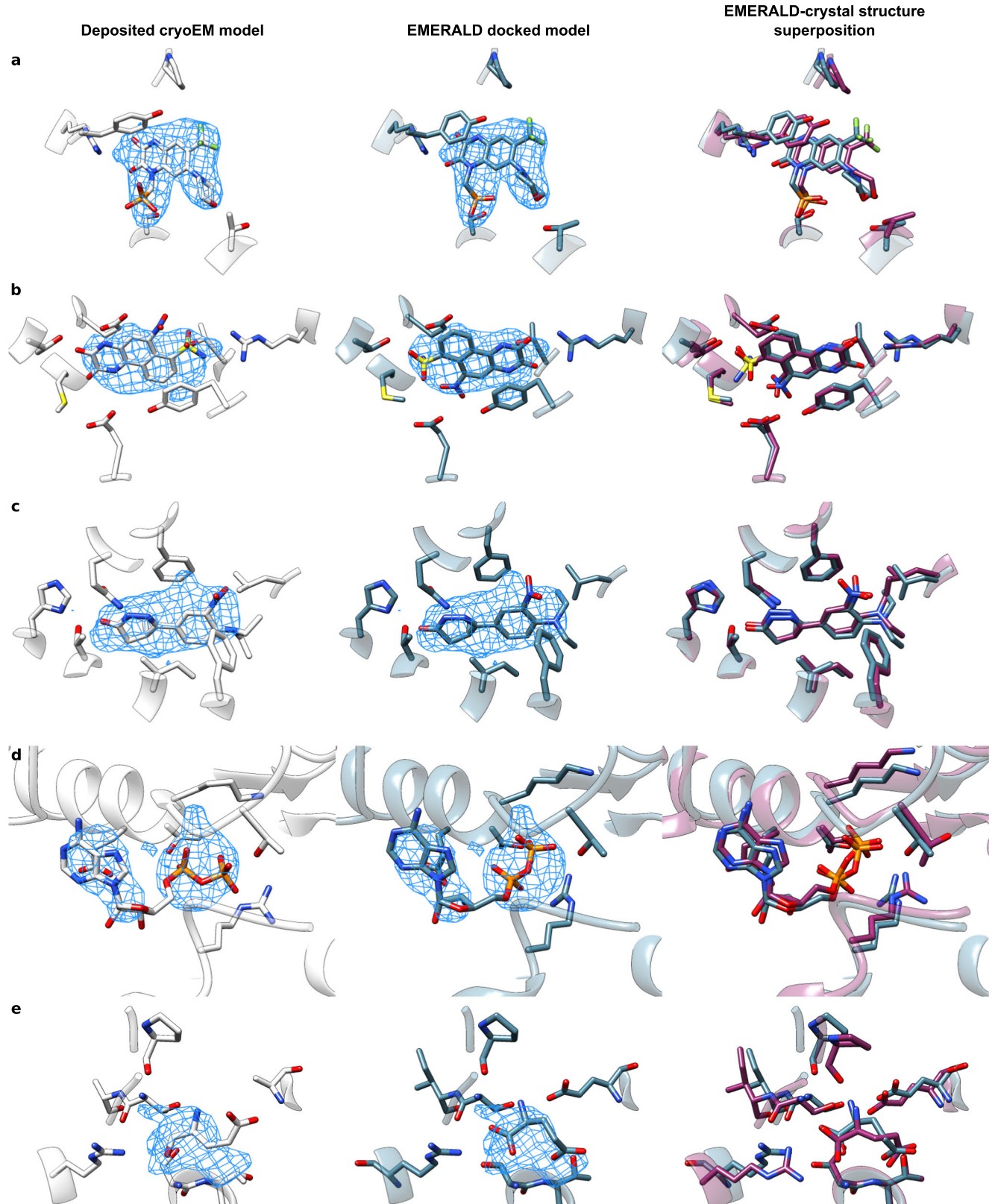

**Fig. 3 | Alternate conformations predicted with EMERALD match high-resolution crystal structures. a–e** Comparison of the deposited model (left, white), EMERALD model (center and right, blue), and higher resolution crystal model (right, purple). **a** Antagonist ZK200775 in AMPA receptor (EMDB: 23292, PDB: 7LEP, local resolution: 3.45 Å) and its associated crystal model (PDB: 5ZG2). **b** Molecule NBQX bound to the AMPA receptor (EMDB: 12805, PDB: 7OCE, local resolution: 2.75 Å) and its associated crystal model (PDB: 6FQH). **c** DNMDP bound to the SLFN12-PDE3A complex (EMDB: 23495, PDB: 7LRD, local resolution: 2.95 Å) and its associated crystal model (PDB: 7KWE). **d** ADP bound to ClpB (EMDB: 21553, PDB: 6W6E, local resolution: 4.42 Å) and its associated crystal model (PDB: 5LJ8). **e** Glutamate ligand in an AMPA receptor (EMDB: 12806, PDB: 7OCF, local resolution: 4.26 Å) and its associated crystal model (PDB: 3TKD).

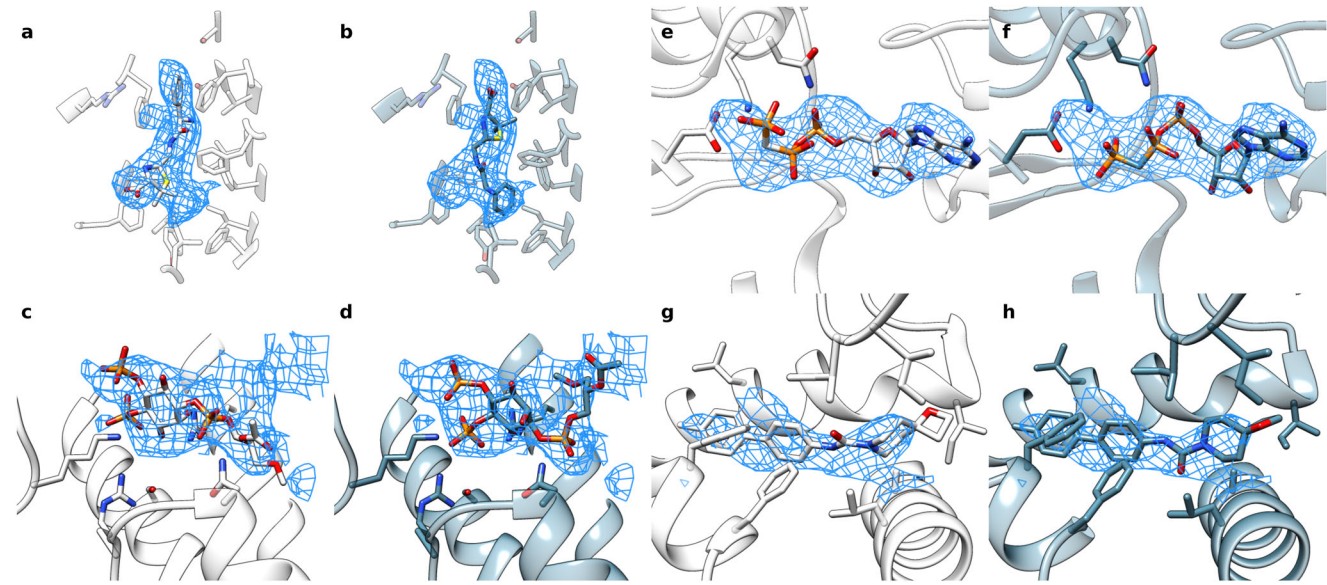

**Fig. 4 | Alternate conformations found by EMERALD in cases without crystal structures. a, b** The deposited structure (**a**) and EMERALD docked model (**b**) of Ampicillin bound to Mtr pump (EMDB: 21228, PDB: 6VKS, local resolution: 3.84 Å). **c, d** The deposited structure (**c**) and EMERALD docked model (**d**) of PIP₂ bound to the TRMP8 ion channel (EMDB: 0487, PDB: 6NR2, local resolution: 4.07 Å). **e, f** The deposited model (**e**) and EMERALD model (**f**) of an ATP analog in flippase ATP11C (EMDB: 30163, PDB: 7BSP, local resolution: 4.06 Å). **g, h** The deposited model (**g**) and EMERALD model (**h**) of GO52 bound to the CD4-HIV-1 Env SOSIP complex (EMDB: 22049, PDB: 6X5C, local resolution: 4.11 Å).

groups of our docked model make more interactions with basic residues that bind PIP₂ in other structures of TRPM8[32]. While the start of the acyl chains are oriented away from the transmembrane region of the protein, this is likely occurring because the chains are truncated. Considering the phosphate placements in the deposited model do not appear in the top 20 lowest-energy models for any trajectory and the reasons above, EMERALD predicts a more accurate model of PIP₂ binding.

Additional cases with confident alternative models are shown in Fig. 4e–h. For the ATP analog in a structure of the ATP11C flippase[33] the gamma phosphate sticks out of density in the deposited model (Fig. 4e) but is modeled into the density and interacting with a nearby lysine residue in the docked model (Fig. 4f). Finally, our EMERALD model of a small molecule GO52 bound to the CD4-HIV-1 Env SOSIP complex[34] confidently fits the amide and piperidine groups into the map better than the deposited map, while keeping the hydrophobic interactions as the deposited model (Fig. 4g, h).

We next identified cases where: (a) the EMERALD model and deposited structure were different, and (b) half maps were available in the EMDB. For these cases, models were refined into one half map and validated against the other using real-space density correlation. When comparing the deposited and EMERALD models (Supplementary Fig. 3a), we found two instances where EMERALD's model fits the validation map worse (Supplementary Fig. 3b–e), seven cases where it fits the validation map better (one of which is shown in Fig. 4d), and saw equivalent quality for the remaining 53 cases.

### Low-confidence unmatched cases show pseudo-symmetry or weak density
While our analysis confidently discovers alternate ligand models, 58% of docked molecules with similar quality to the deposited model have medium or low confidence. We found that small molecules that have pseudo-symmetry or have flexible moieties represent these low-confidence cases because of the challenges they provide from their often noisy and inconclusive density. In some instances, two or more replicates of EMERALD agree on a substructure of the molecule (dark blue, Fig. 5a, b)[35], but differ in a rotamer of a functional group or a

flexible group (light blue, Fig. 5a, b). For other ligands, ambiguous density leads to little agreement among the reference model and low-energy Rosetta models (Fig. 5c, d). The authors for the allosteric modulator of a dopamine receptor note the lack of confidence in the deposited structure[36], but have mutagenesis studies to confirm the conformation modeled (Fig. 5c)[37]. However, one model found with EMERALD aligns with their opposing model and fulfills an unexplained region of density in the deposited model (Fig. 5d). Altogether, these entries show the difficulty in interpreting cryoEM data at medium to low resolution leading to ambiguous density explanations for a single map, and the limits to automated ligand docking using our protocol.

### Cases with worse ligand models show poor initial sampling
To learn what improvements could be made to EMERALD in the future, we looked at instances where EMERALD predicts a ligand with worse metrics than the reference model. We found that these cases often had density that is discontinuous or noisy, leading to incorrect skeletonization. For a ubiquinone binding electron transport protein[38], the density skeleton only finds density near the head group (Supplementary Fig. 4c). Without a complete skeleton, the initial population struggles to find the deposited conformation, placing the head group exposed to solvent (Supplementary Fig. 4b). In this case, if the 2.63 Å data is instead truncated at 4.0 Å resolution, the density becomes more continuous, and the skeleton generated by EMERALD matches the ligand conformation much more closely (Supplementary Fig. 4d). With a complete skeleton, the docked model is no longer worse than the deposited model. The head group of the lowest-energy model makes the same hydrogen bond interactions as the deposited model, and the docked model improves density correlation by 0.03 (Supplementary Fig. 4e). This underscores the importance of the initial sampling step, especially when evaluating ligands with a large number of rotatable bonds, and identifies areas for future upgrades in EMERALD.

### Blind modeling of linoleic acid
To demonstrate our protocol's utility in structure determination, we used EMERALD to create a model for linoleic acid bound in a previously undetermined protein structure. Determining this model

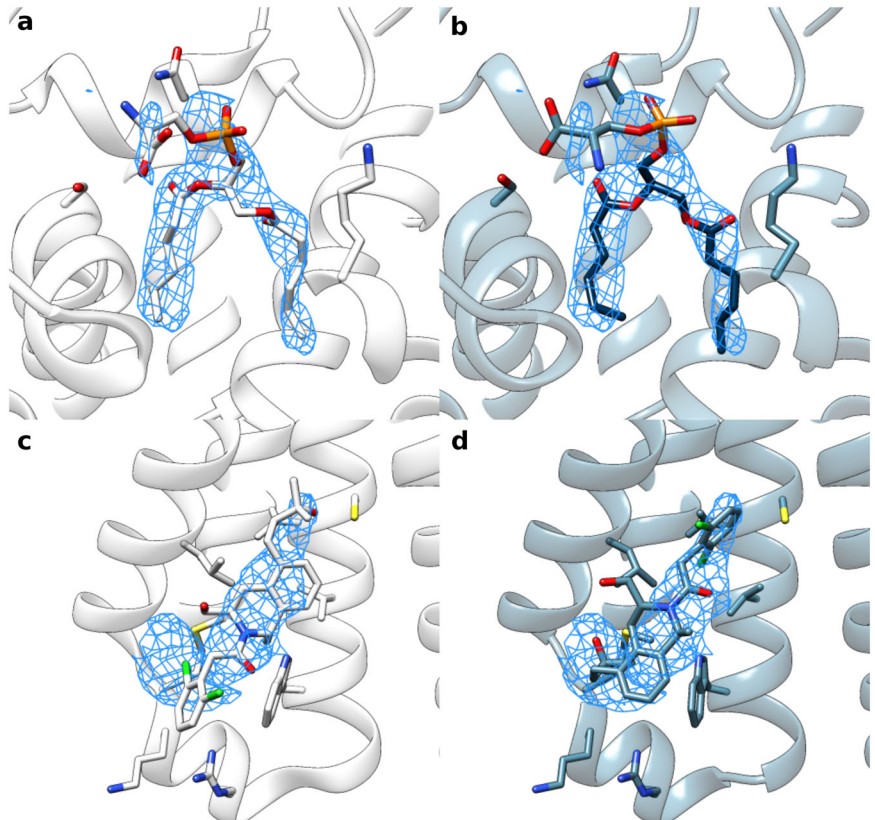

**Fig. 5 | Examples of low-confidence docked models where there may be ambiguity or heterogeneity in the data. a** The deposited model of a phosphoserine lipid in ATPase (EMDB: 21844, PDB: 6WLW, local resolution: 2.75 Å) and the EMERALD-docked model (**b**) place the fatty acid tails in strong density, but have differences in the head group. The lowest-energy models for all three triplicates find the same lipid tail orientations (dark blue). **c** The deposited model of LY3154207 bound to DRD1 (EMDB: 30395, PDB: 7CKZ, local resolution: 3.09 Å) and EMERALD-docked model (**d**) adopt different conformations. With low-resolution density and few residue binding partners, both models are equally plausible.

manually would be an arduous task considering high flexibility of the ligand (Fig. 6a). Despite the difficulty of modeling the suspected ligand, EMERALD predicts a small molecule conformation that fits the density, makes an anchoring electrostatic interaction with a neighboring arginine residue, and introduces little torsional strain throughout the hydrophobic tail (Fig. 6b). This placement is supported by the structure of linoleic acid bound to a related protein[39]. Creating the model required no user input once ligand restraint files were made, and the ease and accuracy when modeling linoleic acid prove the value of EMERALD for structure determination.

## Discussion

Here, we show a method EMERALD that is capable of accurately and automatically producing deposition-ready small molecule models into cryoEM maps without human bias during modeling. After being benchmarked on over one thousand ligand-bound entries in the EMDB, EMERALD identifies a confident solution in 62% of entries, in some cases identifying alternate models supported by crystal structures and map validation. Moreover, we show this fully automated protocol determining the conformation of linoleic acid in a previously unsolved structure.

The method should be generally applicable to most ligands with fewer than 25 rotatable bonds; larger ligands have too large of a search space for this algorithm to effectively sample. Discontinuous or noisy density also proved challenging, though modified map processing to improve density connectivity was shown to rescue at least one of these cases. Our current approach only models a single ligand at a time, which complicates density assignment for structures with ligands close together like electron transport proteins. Finally, Rosetta's poor

handling of metal ions precludes modeling ions as cofactors or as ligands themselves, leaving a significant group of proteins unanalyzed[40]

Currently, our method requires the modeler to know the identification and approximate binding location of the ligand, a non-trivial task when studying novel protein-ligand complexes. For more utility during model building, expanding our method to recognize potential unmodeled ligand blobs and quickly assess possible ligands to determine identity would be beneficial. As is, however, EMERALD offers an automatic tool for ligand modeling that will prove helpful for the now common scenario of ligand-bound structure determination through cryoEM, and EMERALD will serve as a valuable addition to the toolkit of Rosetta EM modeling methods[4,41,42] for model building under one software package.

## Methods

### Creating the protein-ligand dataset

All single-particle EMDB entries with an associated ligand bound structure at 6 Å nominal resolution or better as of September 03, 2021 were obtained. Given the specificity of trying to model ions and glycans, structures with only these types of ligands were excluded from the dataset. In addition, the set had several cases with small molecules in close proximity. To simplify the docking situation, entries with two or more ligands within the binding pocket as defined in our docking protocol were also eliminated from the set. To only have entries with complete macromolecule-ligand complex models that fit the EM density well, structures with a density correlation below 0.4 or that left large regions of density unmodeled were dropped. When considering the first instance of a unique ligand for each EMDB entry, there were a total of 1704 total cases to process for docking.

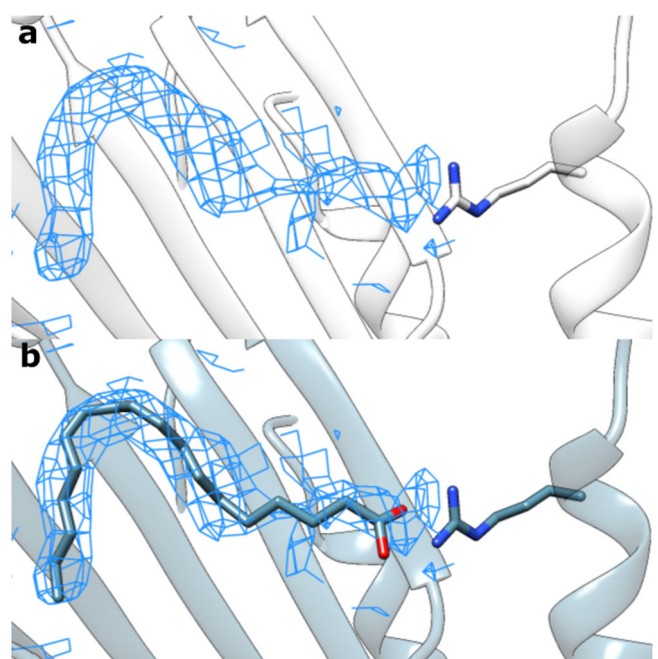

**Fig. 6 | Blind modeling of linoleic acid. a** Unmodeled density for linoleic acid (local resolution: 2.9 Å). **b** Output model from density-guided docking. The model makes an anchoring electrostatic interaction with a nearby arginine residue and models the tail strain-free into the density.

## File preparation for docking

For accurate ligand docking, small molecules need proper protonation states and partial charges. However, the protonation state assigned can depend on the protonation assignment method. To determine the most likely protonation state for a small molecule, we calculated protonation states with three assignment tools—phenix.elbow[43], openbabel[44], and dimorphite[45]—and selected the protonation state assigned with two or more methods. If there was no agreement or failures during the assignment, the phenix.elbow assignment was used for modeling. SDF files of the first instance of each unique ligand-entry pair were downloaded from the PDB and used for input. For processing with phenix.elbow, all possible hydrogen atoms were added to the SDF file using openbabel, and then hydrogen atoms were removed to the final protonation assignment using phenix.elbow. To generate the protonation state with openbabel, the hydrogen atoms were simply added to the downloaded SDF file at a pH of 7.4. Instead of adding protons to a structure, dimorphite (which utilizes RDKit[46]) protonated small molecules as SMILES strings, which were then converted to a structure via openbabel. All three protonation assignment methods agreed for 794 instances with two methods agreeing for 157 cases (Supplementary Fig. 5).

With the protonation state assigned, a mol2 file with AM1-BCC partial charges was generated with antechamber[47,48]. Finally, a Rosetta specific parameters file was created for each ligand. Receptors were cleaned by eliminating non-macromolecular atoms in the PDB file and replacing modified residues with their unmodified correspondent. The ligand to be docked was added to its position in the deposited structure and randomly translated 0.0–2.0 Å in any direction before docking.

## Density erosion and alignment

To ensure the quality of ligand conformations in the ligand pool during the genetic algorithm, randomly perturbed ligands were aligned into unmodeled density to generate the initial pool. Voxels in the density map within 10 Å of the center of mass of the ligand but >2.5 Å from an atom in the receptor were searched and eroded in a modified erosion algorithm from previously described methods[12,49]. Briefly, voxels were

labeled as 0 if their associated density map value was below a density value threshold and labeled as 1 if above it. The voxels were searched in order of density value, and voxels with neighboring voxels of 0 value were removed from the skeleton. If removing a voxel breaks skeleton continuity or if all of the voxel's neighbors had a value of 0, then the voxel was added back into the skeleton. This process was repeated until a skeleton remained of voxels with high density values.

Blobs of density are often discontinuous and difficult to separate from noise at lower resolution. To account for the low resolution, skeletonizing density was performed in two successive steps with increasing strictness on erosion. On the first pass, peaks in the density were detected and eroded only considering voxels sharing a face with each other. This keeps connections between density blobs that may be disjointed. The remaining voxels were clustered into potential skeleton networks by separating groups of voxels that are 3 Å away from another group. Only the largest network of voxels was chosen for further erosion to eliminate noisy voxels. The largest group of voxels underwent a second, stricter erosion that considered all voxels that share a face or edge with each other, leading to a pseudo-atomic skeleton.

The skeleton was used during initial ligand conformer generation of the genetic algorithm to ensure a starting pool of ligands that already fit into the density. Small molecules were randomly translated and perturbed in the binding pocket and half of the small molecules in the initial pool were aligned to the skeleton. For alignment, the ligands were centered on the center of mass of the skeleton, and then atom-skeleton point pairs were determined. The shortest distance of an atom-skeleton pair while searching over all pairs was found, and this search was repeated until either all atoms or all skeleton points had a unique pairing. For the coordinates in each atom-skeleton pair, the topped out harmonic function in Eq. (1) was used to restrain ligand atoms:

$$E_{ij} = 36(1 - e^{-x_{ij}^2/9}) \tag{1}$$

where $E_{ij}$ is an energy penalty applied and $x$ is the distance in Angstroms between the atom-skeleton pair $i, j$. The ligand is aligned into the density over two stages of energy minimization with 20 and 15 short rounds of minimizations with the atom-skeleton restraints updated after each round.

## Docking protocol and analysis

An initial population of 100 ligands were generated by randomly perturbing across a six-fold axis and the torsion angles of the ligand to be docked. Half of the initial ligands were aligned to the density as described above, while the other half of the population were selected from the top 50 models of 5000 random ligand conformations to ensure diversity in the initial population. The population initialization contributes the longest to EMERALD's completion time. All side chains within 5 Å plus the radius of the ligand of the initial ligand center of mass were also considered for optimization. The ligand population and nearby side chains were optimized over 10 generations of a genetic algorithm using default parameters in GALigandDock and a scoring function with a high electron density score weight of 100 to evaluate a ligand's fit into density. The top 20 ligand conformers at the end of the GA were further optimized along with nearby macromolecule atoms using a cartesian minimization in Rosetta. Example scripts for running density-guided ligand docking are provided below.

All entries were run in triplicate and the lowest-energy model for each individual run was further analyzed for docking success. Only cases with 25 or fewer torsion angles were analyzed as the search space of ligands with more torsions becomes difficult to fully explore during a GA. This, along with losing cases from inherent failure during ligand processing, left 1053 cases to analyze. Because of a low confidence in the reference models due to their low resolution, docked models were not directly compared to their respective reference models. Instead, all reference models were relaxed into their EM density map in Rosetta

using the cartesian minimization used after the genetic algorithm. Along with a symmetry-independent RMSD value, docked models were compared to reference models by the number of residues that make hydrogen bonds with the ligand and a density correlation calculated in Rosetta. These metrics were used to categorize docking results as matches (docked pose within 1 Å of relaxed reference model); non-match, similar quality (>1 Å RMSD, density correlation$_{dock}$−density correlation$_{deposited}$ > 0.025 and hydrogen bonds$_{dock}$−hydrogen bonds$_{deposited}$ > −1); or non-match, worse quality (>1 Å RMSD, density correlation$_{dock}$−density correlation$_{deposited}$ < −0.025 or hydrogen bonds$_{dock}$−hydrogen bonds$_{deposited}$ < −1). Further support for docking success was calculated by determining the convergence of lowest energy ligand models across the triplicate runs. The distance between atom pairs across models were calculated and results were further divided into those with two or more trajectories having their lowest energy models within 1 Å RMSD, more than within 1 Å for 60% of atoms, or within 1 Å for fewer than 60% of atoms.

The resolution of cryoEM maps often varies from the nominal resolution of a map, so to analyze the performance of EMERALD against map resolution, we compared docking results to local resolution rather than nominal resolution. Maps with local resolution calculations were generated with MonoRes via the Xmipp software package[50]. The deposited maps were filtered with a Gaussian kernel with a sigma of 0.02 times the map dimensions. Binary masks were created using the filtered maps by keeping voxels with a value above 0.05 times the maximum voxel value in the filtered map. With the binary masks, local resolution estimate maps for all instances were created. To calculate the local resolution surrounding the modeled ligand, the local resolution of all voxels within 5 Å of the ligand were averaged. Voxels with zero local resolution values were not included in the average. Considering that ligand binding sites are often less-resolved areas of a map, the nominal resolution was reported if the calculated local resolution of a map around the ligand was better than 1 Å than the nominal resolution since an error likely occurred.

The following command in Rosetta was used for the low-pass filter of map EMDB-30475:

```
$ROSETTA/main/source/bin/density_tools.default.
linuxgccrelease -truncate_hires 4.0 -mapfile emd_30475.
map -truncate_map
```

### Comparison of docked and EM models to crystal structures
For each ligand-protein pair in the EMDB dataset, the PDB was searched for structures solved by X-ray crystallography at 2.6 Å resolution or better with at least 50% sequence identity to the protein and containing the same ligand. Results from the PDB were filtered further to only contain entries with similar ligand binding pockets as the corresponding EM model. The crystal models were aligned to the EM models by aligning all residues within 10 Å of the ligand using match-maker in UCSF Chimera[51]. Once aligned, the density correlations of the ligands in the crystal models were calculated in Rosetta. All entries with a pocket-aligned RMSD > 1.5 Å and a ligand density correlation lower than 0.1 of the EM model were discarded for being too unalike. This gave 129 ligand-bound EMDB structures with similar crystal models. The 100 cases from this set where EMERALD converged on the same model were analyzed by RMSD to the ligand in the aligned crystal model.

### Half-map validation of docked models
Half maps for the instances with the non-match, similar or better quality designation and EMERALD convergence were obtained from the EMDB when available, giving 62 cases. Maps were sharpened with phenix.auto_sharpen[52], and the deposited and docked models were refined into the first half map using a dualspace refinement in Rosetta.

The density cross correlations for both ligand refined models were calculated with the first and second half map. If the difference in density correlation with the second map was >0.05 between the two models, the model with the higher correlation was considered better.

### Visual analysis and images
Figures of ligand-bound models and their EM maps were created using UCSF Chimera[53]. Maps displayed in figures were changed to 1.0 Å using the vop command in Chimera for visual consistency. Plotting of data was performed using the ggplot2 package in R[54].

### Example EMERALD scripts
Example Rosetta XML script for docking:

```
<ROSETTASCRIPTS>
<SCOREFXNS>
   <ScoreFunction  Name="relaxscore"  weights="beta_gen
pot">
<Reweight scoretype="elec_dens_fast" Weight="100">
<Reweight scoretype="gen_bonded" Weight="1.0">
<Reweight scoretype="coordinate_constraint" Weight="1.
0">
</ScoreFunction>
</SCOREFXNS>
<MOVERS>
<SetupForDensityScoring Name="setupdens" >
<LoadDensityMap Name="loaddens" mapfile="%%map%%" >
<GALigandDock Name="dock" scorefxn="relaxscore" ngen=
"10" npool="100" rmsdthreshold="1.0" smoothing="0.0"
ramp_schedule="0.1,1.0" grid_step="0.325" padding="5.
0" nativepdb="%%native%%" sidechains="auto" final_exact_
minimize="bbsc"  random_oversample="100"  use_pharmaco
phore="false" skeleton_threshold_const="5.0" neighbor
hood_size="7"  sample_ring_conformers="1"  reference_
pool="map">
</MOVERS>
<PROTOCOLS>
<Add mover="setupdens">
<Add mover="loaddens">
<Add mover="dock">
</PROTOCOLS>
<OUTPUT scorefxn="relaxscore">
</ROSETTASCRIPTS>
```

Example command line for docking

```
$ROSETTA/main/source/bin/rosetta_scripts.linuxgc
crelease \
-in:file:extra_res_fa $ligand_params_file \
-in:file:overwrite_database_params \
-gen_potential \
-database $ROSETTA/main/database \
-score::gen_bonded_params_file scoring/score_functions/
generic_potential/generic_bonded.round6p.txt \
-s $input_model \
-overwrite \
-multi_cool_annealer 10 \
-parser:protocol $xml:file \
-parser:script_vars map=$em_map \
-atom_mask 2 \
-sliding_window 1 \
-edensity::score_sliding_window_context \
-edensity::mapreso $reso \
-edensity::grid_spacing 1.0 \
-no_autogen_cart_improper
```

## Reporting summary

Further information on research design is available in the Nature Portfolio Reporting Summary linked to this article.

## Data availability

The data that support this study are available from the corresponding authors upon reasonable request. PDB accession codes used in this manuscript are: 7LEP, 5ZG2, 7OCE, 6FQH, 7LRD, 7KWE, 6W6E, 5LJ8, 7OCF, 3TKD, 6VKS, 6NR2, 7BSP, 6X5C, 6WLW, 7CKZ, 7M5E, 5VYH, 7OJ8, 6UKJ, 7CUW. EMDB accession codes used in this manuscript are: 23292, 12805, 23495, 21553, 12806, 21228, 0487, 30163, 22049, 21844, 30395, 23674, 12939, 20806, 30475. Models with hydrogen atoms for EMERALD-docked models in all main figures (Figs. 3–6) are provided in Supplementary Data 1. Source data for Figs. 2B, C, and Supplementary Fig. 3A are provided with the paper. The lowest energy models for all cases for each individual EMERALD run are available for download at https://files.ipd.uw.edu/pub/EMERALD/EMERALD_top1_models.tar.gz [https://files.ipd.uw.edu/pub/EMERALD/EMERALD_top1_models.tar.gz]. Source data are provided with this paper.

## Code availability

All methods described are available as part of Rosetta, using weekly releases after February 5, 2023 (version 2023.06 or later). The Rosetta XML files and flags for running all the refinements discussed in this manuscript are included in Methods. A demo for running EMERALD is included in Supplementary Data 2.

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

## Acknowledgements
This study was supported by the National Science Foundation Graduate Research Fellowship under Grant No. DGE-1762114 (to A.M.), a grant from Thermo Fisher Scientific (to A.M.), the National Institute of General Medical Sciences (1R01GM123089-01, to F.D; T32GM008268-32 to S.K.Z.), the Defense Threat Reduction Agency (GRANT13030960, to F.D. and G.Z.), the National Institute of Allergy and Infectious Diseases (F31AI174573-01 to S.K.Z.; DP1AI158186 and 75N93022C00036 to D.V.), a Pew Biomedical Scholars Award (D.V.), an Investigators in the Pathogenesis of Infectious Disease Awards from the Burroughs Wellcome Fund (D.V.), Fast Grants (D.V.), the Bill & Melinda Gates Foundation (OPP1156262 to D.V.), the University of Washington Arnold and Mabel Beckman cryoEM center and the National Institute of Health grant S10OD032290 (to D.V.). D.V. is an Investigator of the Howard Hughes Medical Institute.

## Author contributions
A.M. and F.D. conceived the study, designed the experiments, and analyzed the results. G.Z. developed methodology used in data analysis. S.Z. and D.V. collected and analyzed cryoEM data. All authors wrote the manuscript.

## Competing interests
The authors declare no competing interests.
