## [Peer Review File · Nature Communications]

Automatic and accurate ligand structure determination guided by cryo-electron microscopy mapsReviewers' Comments:

Reviewer #1:

Remarks to the Author:

This manuscript presents a new computational approach for determining structures of ligands bound to proteins from an electron density map determined using cryo-EM. This method will be a worthwhile addition to the ROSETTA suite. While the method seems effective, I have some reservations about how broad its impact will be. There are already mature tools for aiding users in determining protein structures from cryo-EM maps. Although this requires a significant level of expertise of the user, any institution with a cryo-EM facility certainly has staff who are skilled at this. In comparison to this task, assigning a reasonable geometry of a known ligand to a cryo-EM structure where the protein structure has already been solved is relatively easy task, so EMERALD is not likely to change the way these structures are solved. The researchers highlight some cases where EMERALD seems to do better than the published structure, but this is clearly not the norm because most cryo-EM groups are adept at determining these structures already and a more careful structure assignment would have fixed these issues. The examples that EMERALD fixes could have been fixed a diligent scientist - and there is no way of determining a good quality structure from cryo-EM structure without a diligent scientist.

Along these lines, reading the supporting information shows that this method is far from a black-box - so it is not obvious that this will supplant the existing approaches. The experimental resolution of cryo-EM is rapidly improving, so this kind of cryo-EM guiding docking will only be decisive in progressively narrower set of structures going forward.

In terms of methods, the only part where I have any serious criticism is the step where pseudo-atomic centers are designated and then there is an attempt to align the ligand to them. It appears to work fairly well, but I find the introduction of the pseudoatoms to be inelegant. The SI reveals that predicting binding is sensitive to this skeletonising and there was a lot of work done to generate reasonable pseudoatoms (e.g., erosion algorithms, etc). Is there not a way to align the ligands with the density map directly? The rest of the approach is generally reasonable.

My only other quibble is with the use of PDB structures to assess their algorithm. These are approximate models too, typically constructed using a forcefield-based crystal structure solving program. An ideal cryo-EM protein-ligand method would do better than these methods, not the same as them. The authors match cryo-EM structures to ligands in homologous proteins. This successful to a point, but the two poses should not be in perfect agreement because the fine structure of the conformation of the ligand should be somewhat distinct in a protein with only moderate homology. This comparison just confirms that their algorithm generates similar poses to those solved using traditional approaches. Ideally, the comparison would be made to the cryo-EM density directly. Admittedly, many of the maps are likely too coarse to be of much use for this purpose, but that's just it - there may be no way of determining experimentally how well this method is really doing using the data on hand.

I would also have liked to see some metrics on which structures were excluded. This seems likely to select structures with well ordered structures. This method is far more interesting and useful in the cases that this comparison leaves out - structures with limited resolution where there is no homologous solution. It is not clear to me that the test set used is applicable to these cases.

Reviewer #2:

Remarks to the Author:

The authors present an automated tool for modeling of a protein-ligand complex into cryoEM maps with ROSETTA-based tools. While the concept is timely, interesting, and I think would bring a lot to the discussion of modeling ligands into cryoEM maps, unfortunately the overall validity of the work is

difficult to assess without PDB files with hydrogens for at least the poses that are explicitly depicted in figure panels, as I will elaborate on further. With this information in hand I think the manuscript can be further reviewed, however without the files provided I would not suggest publication.

One of the largest potential problems that needs to be assessed in any ligand modeling task, especially an automated one, is how tautomerization and protonation states are assigned. This is recognized in the method section, however to my knowledge the proposed solution of phenix.elbow only retains the protonation state provided by the modeler in the default mode (or, with some flags provides extremely crude adjustments). This would be an advantage of other automated cryoEM ligand modeling tools that can perform actual calculations for this task.

Assigning protonation deserves more discussion, as accurate protonations matter to such a degree that I suspect this has introduced errors into several of the results. I would specifically like to call into question Figure 4h; I would not think this is an equally valid alternative conformation, I think this pose is almost certainly incorrect. The dopamine receptor is an aminergic GPCR, their ligand recognition in the vast majority of cases involves a salt bridge between a protonated amine and an aspartate on the receptor. This is present in the original pose but absent in the EMERALD pose, which is described as having no substantive differences in interactions. I suspect the ligand was simply not correctly protonated, causing both the docking algorithm and the assessment of pose quality to miss the lack of this key interaction. (I would also point out just by eyeballing the original pose looks like a substantially better fit to the map, but again this is where having the PDB files would significantly help the assessment of the manuscript). This would lead me to wonder how many of the 'equally valid alternative poses' correspond to incorrect protonation events. It is possible that using the correct state of the ligands will yield improved (or even substantially improved) performance for EMERALD.

A smaller issue:

Line 46: MDFF is not to my knowledge built around glide.

I hope to have the opportunity to review the manuscript further as I do think it would be a very interesting addition to the field, but I would need the necessary data for review.

Reviewer #3:

Remarks to the Author:

Automatic and accurate ligand structure determination guided by cryo-electron microscopy maps ^{SEP}

Andrew Muenks, Samantha Zepeda, Guangfeng Zhou, David Veessler, Frank DiMaio^{SEP}

The manuscript presents a tool EMERALD for fitting/docking ligands into cryoEM reconstructions, with a focus on practical application and demonstrating performance using a test set taken from the EMDB. The manuscript is organised and written well, and the results are positive, indicating success of the method and implementation - undoubtedly the tool will be useful and a welcome addition to the community's arsenal of cryoEM model building and analysis tools. However, there are a number of issues that should be considered, primarily pertaining to: (1) the presentation of the results relating performance to relevant factors such as resolution and ligand size; (2) referring to overall rather than local resolution in the context of localised fitting/docking; and (3) limited approaches to validation of the presented results. There are also a number of statements that are inappropriate or unsupported by the evidence presented. I would recommend reconsideration for publication following revision of the manuscript after considering the following comments.

Ligand docking success as a function of resolution and ligand size (proxy: number of torsions) is presented in Supp Fig 2, but largely glossed over in the main text (briefly mentioned on Page 3 line 99). I feel that this type of analysis is of high relevance and interest to the prospective reader when

considering the overall performance and suitability for application in different scenarios. The authors should consider improving the quality of this analysis (as per comments below), and elevating it to being presented in figures in the main article rather than as Supplementary Information. Indeed, most figures selected for inclusion in the main article present specific example cases (Figures 3-6), yet there is only one limited figure to illustrate overall performance via large scale analysis (Figure 2), so I feel an additional figure relating performance to relevant factors would be justified/appropriate.

In Supp Fig 2 docking success is analysed as a function of map resolution. This is a very important analysis, but the way it is presented is misleading due to using the nominal resolution of the overall map rather than the local resolution of the map in the vicinity of the ligand. The result is that the dependency of docking success on map resolution is most likely highly underestimated. The authors conclude that "docking success is resilient to changes in overall map resolution" (Page 3 line 98), which I feel is very misleading, and quite possibly inaccurate. Ideally the authors should use local resolution for such analyses - there are various tools available for calculating estimates of local resolution (e.g. ResMap).

Still looking at Supp Fig 2, the decision to show density rather than frequency on the vertical axes is understandable. However, the frequencies should be made available so that the subfigures can be more fairly compared. I would suggest adding the numeric frequencies to the plots themselves, using the same green/orange/blue colours. Or alternatively switch to showing frequencies on the vertical axes. It may also be worth considering whether violin plots would be an informative representation.

I would suggest also adding additional complementary visuals that will allow better analysis of the effect of resolution and number of torsions on docking success. I would say that the quantity of interest is the success rate - i.e. the proportion of successes versus failures - so this should be the focal quantity shown on the vertical axis. If appropriate, differences in sample size for different resolutions/torsions can be accounted for by adding confidence intervals based on standard error.

It seems that there are very few cases below 5 Å resolution, and of those there seem to be more cases where EMERALD fails than cases where it correctly docks the ligand. Consequently, the statement in the Abstract that "this method is robust at predicting ligands in maps as low as 6Å resolution" is not supported by the results presented. I note that the novel example presented in Figure 6 fits linoleic acid into a cryoEM map with nominal resolution 2.85 Å. This issue of resolution is further exacerbated by confusing the overall nominal resolution of a map, and map resolution in the context of ligand docking (i.e. local resolution). Hopefully rethinking the presentation of the results (as per my comments above) will facilitate the ability to make more precise statements that are backed up by appropriate levels of supporting evidence.

The referencing of other works in the Introduction could be improved. Some of the references for tools originally developed for macromolecular crystallography (MX) (references 10-14) are relevant but quite old (2001-2007). Some of these programs have in more recent years been adapted for cryoEM - this should at least be acknowledged, citing the corresponding more recent papers (e.g. Casanal, Lohcamp & Emsley, 2019; Chojnowski, Sobolev, Heuser & Lamzin, 2021; etc.), irrespective of whether the authors feel that those tools are unsuited to application at more moderate cryoEM resolutions. Note that the authors' use of nominal overall rather than local resolution makes such judgements about the suitability of different tools at different resolutions ill-defined. Also, since there are no direct comparative performance analyses of different tools in the present manuscript, nor any prior such analyses referenced, I would be very careful about pigeonholing particular tools as being suitable only at certain resolutions without citing due evidence, for fear of unintentionally causing offence via slander! (Did the authors try using other tools to dock the linoleic acid shown in Figure 6?)

I also struggle with the comment: "But as resolution decreases below 3Å, the topological features these methods rely on become less defined, and their accuracy in modeling ligands within 1Å RMSD of a reference ligand falls below 20% [13, 14]". My main problem with this is that the concept of

resolution is completely different in MX (resolution of the highest miller indices selected for use in refinement) and in cryoEM (based on half-map FSC criteria), to the point where the numerical values cannot be directly compared. References 13-14 are from 2006-2007 and refer to application in MX, and thus the quoted conclusions do not transfer directly to cryoEM. Note that the fact that cryoEM "data" include phase information has a positive effect on the quality of the maps relative to MX, in contrast with MX where phase information is lacking and the maps are conditional on the current state of the model (and there are also other relevant differences between MX and cryoEM maps). Indeed, for a given nominal resolution, MX maps tend to be much worse than cryoEM maps overall, meaning that it might be argued that ligand fitting is generally a harder challenge in MX than in cryoEM.

In the context of discussing other tools in the Introduction, it might be worth also acknowledging that the various tools have been developed with different intended use cases in mind - in particular, the tradeoff between computational performance and docking quality. It is quoted that the present tool takes 30-120mins to run (Page 2 line 75), whereas some other available ligand fitting tools might take just a few seconds to run. There is a difference between a quick attempt at identifying reasonable initial conformer (that would require further manual intervention and/or model refinement), versus a more computationally intensive fully automated solution to docking (including refinement of the structural environment). It seems that EMERALD is intended to perform the latter.

That being said, from the "Docking protocol and analysis" section in the Supplementary Material it is not completely clear whether the protein model in the vicinity of the ligand is also refined in order to optimise the protein-ligand stereochemical interactions, or whether the ligand is simply docked without modifying the protein model - this should be clarified, adding more methodological/implementation details.

One weakness is that I see no mention of half-maps. Is there a way that half maps can be used to complement the approach, for validation / assessment of confidence in the docked ligand model?

In the cases of ambiguous maps (e.g. Figure 5), did the authors consider using complementary maps in order to attempt to further critique the docked ligand models? For example, locally sharpened maps (e.g. LocScale) and difference maps (e.g. Servalcat); there may be other similar tools for model/map analysis.

On pages 4 and 5, in reference to Figures 3 and 4, it is not sufficiently clear whether the references to finding "alternate conformations" implies that the deposited models are wrong and that the EMERALD-docked ligands are correct, or whether it is implied that both the deposited and EMERALD-docked ligands are correct / plausible conformations. The authors' intended interpretation should be clarified, just to make this completely explicit for the reader. In this context I will point out the comment in the Discussion on Page 7 line 242: "in some cases correcting errors present in the deposited model".

But for now, I will assume that the authors intend for the reader to understand that the identification of alternate conformations does not necessarily mean that the deposited conformations are wrong, as multiple conformations may actually be present. However, if proposing a multi-conformer binding site model, then appropriate validation should be performed in order to verify that it is better than a single-conformer model. When publishing works such as this, there is a certain responsibility to demonstrate best practice so that potential users can follow suit in practical application; in that sense, the current status of the presented examples (Figures 3 and 4) is a half-finished solution. In order to further verify and model such multi-conformer cases, I would suggest overlaying the conformers, modelling them together by assigning different alt conf IDs, and performing refinement of the binding site (including refinement of occupancies). Only then, by comparing the RSCC values and difference maps, would it be possible to claim that the modelling of the binding site is improved by modelling the ligand in multiple alternate conformations.

Specific comments by page/line:

Page 1 line 7 - Affiliations 1 and 2 appear to be the same.

Page 1 line 22 - It seems strange to refer to "This tool" in the Abstract, without mentioning the name of the tool (EMERALD).

Page 2 line 47 - The comment "However, both require user input in either selecting models" reads like a criticism of those approaches... so is it implied that EMERALD doesn't require the user to select a model?! The Discussion on Page 7 line 255 implies otherwise: "our method requires the modeler to know the identification and approximate binding location of the ligand".

Page 2 line 52 - Inconsistent use of "force field" vs "forcefield".

Page 2 line 57 (and elsewhere) - I would suggest using the word "map" rather than "density" throughout the manuscript, given that maps output from cryoEM reconstructions correspond to electrostatic potential, not (electron) density.

Page 2 line 75 - It is good to give an indication of computation time (30-120 minutes). However, it would be interesting to know how that time is partitioned into the different steps? Is there one step that dominates the computation time? Is overall time essentially determined by ligand size, or is the protocol sensitive to decisions/parameters such as the number of conformers to trial?

Page 3 line 102 - In this particular context, "low resolution" should not be hyphenated. It should either be "low resolution of the maps" or "low-resolution maps".

Page 4 line 127 - I'm confused about the switch from "1Å" to "1.0Å"? Also note that technically there should be a space between the numeric value and the unit (e.g. "1 Å")... I won't list all instances of this.

Page 4 line 136 - "are" should be "our".

Page 4 - inconsistency writing "Figure" or "Fig." when referencing figures.

Page 4 line 142/143 - I don't think it's necessary to refer to Fig 3d twice here.

Page 7 line 257 - I don't think it's appropriate to say "our method could expand to recognize potential unmodeled ligand blobs and quickly assess possible ligands to determine identity", given that it is not clear how the method could be straightforwardly expanded to provide those functionalities. Rather, it is sufficient to simply state that such functionalities would be desirable.

Figure 1 - The caption would benefit from some more verbosity. In particular, in subfigure (a) the red cross and black circle lack explanation, and the blue arrows in subfigure (c) that represent the genetic algorithm are too cryptic.

Figure 2 - "4% were 1Å RMSD of" should be "4% were more than 1Å RMSD from".

Figures 3-6 - Unnecessary duplication between the figure captions and the main text in the Results section. The figure captions should be an objective description of the contents of the subfigures, whereas the main text should provide a more detailed narrative - the overlap between these should be minimised where possible. Also, I do not think it is necessary to provide EMD and PDB codes and citations in both the main text and figure captions - I would lean towards providing these only in the figure captions in order to improve readability of the main text.

Figures 3-6 and Supplementary Figures 3-4 - for all EMD entries used as examples the nominal

resolutions should be stated. This is relevant, in order to give the reader an indication of relative map interpretation difficulty. Providing a measure of local resolution would be even better, for the reasons stated above.

Given that a number of improved (or alternative) ligand poses have been identified during the course of the present work, have the authors reached out to the original depositors of the PDB models in the interest of improving the quality of the models in the database and derived biological interpretations?

*Reviewer #1 (Remarks to the Author):*

*This manuscript presents a new computational approach for determining structures of ligands bound to*
*proteins from an electron density map determined using cryo-EM. This method will be a worthwhile*
*addition to the ROSETTA suite. While the method seems effective, I have some reservations about how*
*broad its impact will be. There are already mature tools for aiding users in determining protein structures*
*from cryo-EM maps. Although this requires a significant level of expertise of the user, any institution with*
*a cryo-EM facility certainly has staff who are skilled at this. In comparison to this task, assigning a*
*reasonable geometry of a known ligand to a cryo-EM structure where the protein structure has already*
*been solved is relatively easy task, so EMERALD is not likely to change the way these structures are*
*solved. The researchers highlight some cases where EMERALD seems to do better than the published*
*structure, but this is clearly not the norm because most cryo-EM groups are*
*adept at determining these structures already and a more careful structure assignment would have fixed*
*these issues. The examples that EMERALD fixes could have been fixed a diligent scientist - and there is*
*no way of determining a good quality structure from cryo-EM structure without a diligent scientist.*

While the reviewer makes some valid points, we disagree with the two main points:

- ● While there are mature tools (like AlphaFold2) that can be very accurate for protein structure
prediction, the problem of ligand determination is not completely straightforward. These tools are
often not able to predict conformational changes upon ligand binding, leading to ambiguity in
sidechain positions at marginal resolutions, which complicates ligand determination
- ● While perhaps the errors highlighted in the manuscript might not have been made by a “diligent
scientist,” the fact that they were made in deposited structures highlights the need for such a tool.
It is clear from the mistakes we identify that such a tool will be useful in analyzing and
determining conformations of ligand-bound structures.

*Along these lines, reading the supporting information shows that this method is far from a black-box - so it*
*is not obvious that this will supplant the existing approaches.*

While the method involves several steps, there is no human intervention required, making it a tool with a
low barrier for entry of novice and experienced users. The protocol described in this manuscript can be
followed by other groups to replicate our results exactly.

*The experimental resolution of cryo-EM is rapidly improving, so this kind of cryo-EM guiding docking will*
*only be decisive in progressively narrower set of structures going forward.*

While this is somewhat true (I'd argue against “rapidly”), even high-resolution cryoEM maps have highly
heterogeneous resolutions, where parts of the map are at low resolution (see our response to reviewer 3).
If these portions happen to be a ligand-binding site of interest, the method outlined in the manuscript will
prove incredibly important, even if the nominal resolution of the map is “high.” For instance, PMID:
31160783, PMID: 31792450 and PMID: 35737812 describe structures at high average resolution with
ligands bound at their surfaces and which are therefore not resolved at the same resolution as the core of
the proteins. This is further supported by Review Fig. 3 below showing that the local ligand resolution is
generally lower than the average map resolution. This method will thus help with all these types of
structures (and there are many of them).

*In terms of methods, the only part were I have any serious criticism is the step where pseudo-atomic*
*centers are designated and then there is an attempt to align the ligand to them. It appears to work fairly*
*well, but I find the introduction of the pseudoatoms to be inelegant. The SI reveals that predicting binding*

*is sensitive to this skeletonising and there was a lot of work done to generate reasonable pseudoatoms*
*(e.g., erosion algorithms, etc). Is there not a way to align the ligands with the density map directly? The*
*rest of the approach is generally reasonable.*

This is a fair point. Skeletonization rather than directly fitting density was chosen largely for speed
reasons, as it allows us to align and minimize a pool of candidate conformations very rapidly; performing
equivalent calculations using density directly would be somewhat slower. The main point of this step,
however, isn't to solve the problem, but to generate a pool of candidate structures with reasonable
density agreement. To that end, the method we have chosen works well.

*My only other quibble is with the use of PDB structures to assess their algorithm. These are approximate*
*models too, typically constructed using a forcefield-based crystal structure solving program. An ideal cryo-*
*EM protein-ligand method would do better than these methods, not the same as them. The authors match*
*cryo-EM structures to ligands in homologous proteins. This successful to a point, but the two poses*
*should not be in perfect agreement because the fine structure of the conformation of the ligand should be*
*somewhat distinct in a protein with only moderate homology. This comparison just confirms that their*
*algorithm generates similar poses to those solved using traditional approaches. Ideally, the comparison*
*would be made to the cryo-EM density directly. Admittedly, many of the maps are likely too coarse to be*
*of much use for this purpose, but that's just it - there may be no way of determining experimentally how*
*well this method is really doing using the data on hand.*

This is also a good point. As the reviewer points out, crystal structures themselves are not perfect
(particularly at moderate resolutions) and there are also likely modest conformational changes between
the crystal and EM data. For that reason, we tried to focus our analysis on cases where there were
significant differences ($>1.5 \text{ \AA}$) between the deposited and EMERALD-predicted structure.

*I would also have liked to see some metrics on which structures were excluded. This seems likely to*
*select structures with well ordered structures. This method is far more interesting and useful in the cases*
*that this comparison leaves out - structures with limited resolution where there is no homologous solution.*
*It is not clear to me that the test set used is applicable to these cases.*

Upon reviewing, the "Creating protein-ligand dataset" methods section was not clear on the exclusion
criteria for a case fitting into the EM map. Cases were excluded if the entire macromolecule-ligand
complex had a poor correlation with the EM map or there were large macromolecule areas unmodeled.
These were excluded because there were 83 cases where the structure was misaligned to the map or
there was unmodeled protein density near the ligand that would get picked up in density detection, but
wouldn't if the macromolecule was modeled properly. The text has been updated to make it clearer that
the sentence is referring to the entire pdb model.

*Reviewer #2 (Remarks to the Author):*

*The authors present an automated tool for modeling of a protein-ligand complex into cryoEM maps with*
*ROSETTA-based tools. While the concept is timely, interesting, and I think would bring a lot to the*
*discussion of modeling ligands into cryoEM maps, unfortunately the overall validity of the work is difficult*
*to assess without PDB files with hydrogens for at least the poses that are explicitly depicted in figure*
*panels, as I will elaborate on further. With this information in hand I think the manuscript can be further*
*reviewed, however without the files provided I would not suggest publication.*

Models with hydrogen atoms for all main panel figures are provided with our revision. We have opted to
exclude hydrogens from the figure panels for clarity of figures, but if the reviewer feels this is necessary, it
can be updated.

*One of the largest potential problems that needs to be assessed in any ligand modeling task, especially*
*an automated one, is how tautomerization and protonation states are assigned. This is recognized in the*
*method section, however to my knowledge the proposed solution of phenix.elbow only retains the*
*protonation state provided by the modeler in the default mode (or, with some flags provides extremely*
*crude adjustments). This would be an advantage of other automated cryoEM ligand modeling tools that*
*can perform actual calculations for this task.*

Agreed. We had found that phenix.elbow was capable of removing hydrogens to the correct protonation
state of a fully protonated input ligand model. However, as discussed below, there were 61 protonation
errors, and we have expanded to using other protonation methods to confirm the protonation states used
or highlight cases with incorrect protonation states. The methods section has been changed to include the
updated protonation protocol.

*Assigning protonation deserves more discussion, as accurate protonations matter to such a degree that I*
*suspect this has introduced errors into several of the results. I would specifically like to call into question*
*Figure 4h; I would not think this is an equally valid alternative conformation, I think this pose is almost*
*certainly incorrect. The dopamine receptor is an aminergic GPCR, their ligand recognition in the vast*
*majority of cases involves a salt bridge between a protonated amine and an aspartate on the receptor.*
*This is present in the original pose but absent in the EMERALD pose, which is described as having no*
*substantive differences in interactions. I suspect the ligand was simply not correctly protonated, causing*
*both the docking algorithm and the assessment of pose quality to miss the lack of this key interaction. (I*
*would also point out just by eyeballing the original pose looks like a substantially better fit to the map, but*
*again this is where having the PDB files would significantly help the assessment of the manuscript). This*
*would lead me to wonder how many of the 'equally valid alternative poses' correspond to incorrect*
*protonation events. It is possible that using the correct state of the ligands will yield improved (or even*
*substantially improved) performance for EMERALD.*

Thank you for correctly pointing out the error in Figure 4h. The nitrogen in our ligand model was not
protonated based on our protonation method. Two other protonation assignment methods, openbabel and
dimorphite, correctly protonate the ligand. When the re-protonated ligand is provided to EMERALD, the
lowest-energy docked models across 3 replicates match the deposited model (Review Fig. 1).

Review Fig. 1. If the nitrogen atom is not protonated, EMERALD (center, blue) disagrees with the
deposited model (left, white). When protonated correctly, EMERALD produces a model similar to the
deposited model (right, yellow).

To see how many other cases have disagreements in protonation state assignment, the protonation
 states for all cases in the manuscript were evaluated with openbabel and dimorphite (Review Fig. 2). For
 794/1053 cases, all 3 methods agreed on the same protonation state, and our original protonation
 assignment was confirmed by only one other method for 96 cases. We re-docked the 61 ligands with a
 different protonation state than originally used and figures and percentages have been updated in the
 manuscript accordingly. The overall docking results changed very little (Review Table 1), but now it is
 likely the correct protonation assignment was used for all cases.

Review Fig. 2. Agreement of protonation state assignments. Three protonation assignment methods were
 used to determine the protonation state for a small molecule in EMERALD. The state assigned by 2 or
 more methods was used for docking.

 Review Table 1. Docking results with corrected protonation states.

Result	Original	Re-protonated
Similar or better quality, EMERALD converged	166	168
Similar or better quality, EMERALD substructure converged	169	170
Similar or better quality, EMERALD did not converge	63	63
Match, EMERALD converged	490	489
Match, EMERALD substructure converged	62	61
Match, EMERALD did not converge	56	54
Worse quality, EMERALD converged	10	11
Worse quality, EMERALD substructure converged	31	31
Worse quality, EMERALD did not converge	6	6

A smaller issue:

Line 46: MDFF is not to my knowledge built around glide.

It is not built around Glide, but the reference cited does use Glide to create initial starting poses for MDFF. The manuscript has been updated to avoid any confusion.

I hope to have the opportunity to review the manuscript further as I do think it would be a very interesting addition to the field, but I would need the necessary data for review.

Thanks again for the great comments and catching the protonation state error. The comments bolstered up the quality of the manuscript.

Reviewer #3 (Remarks to the Author):

Automatic and accurate ligand structure determination guided by cryo-electron microscopy maps

Andrew Muenks, Samantha Zepeda, Guangfeng Zhou, David Veessler, Frank DiMaio

The manuscript presents a tool EMERALD for fitting/docking ligands into cryoEM reconstructions, with a focus on practical application and demonstrating performance using a test set taken from the EMDB. The manuscript is organised and written well, and the results are positive, indicating success of the method and implementation - undoubtedly the tool will be useful and a welcome addition to the community's arsenal of cryoEM model building and analysis tools. However, there are a number of issues that should be considered, primarily pertaining to: (1) the presentation of the results relating performance to relevant factors such as resolution and ligand size; (2) referring to overall rather than local resolution in the context of localised fitting/docking; and (3) limited approaches to validation of the presented results. There are also a number of statements that are inappropriate or unsupported by the evidence presented. I would recommend reconsideration for publication following revision of the manuscript after considering the following comments.

Ligand docking success as a function of resolution and ligand size (proxy: number of torsions) is presented in Supp Fig 2, but largely glossed over in the main text (briefly mentioned on Page 3 line 99). I feel that this type of analysis is of high relevance and interest to the prospective reader when considering the overall performance and suitability for application in different scenarios. The authors should consider improving the quality of this analysis (as per comments below), and elevating it to being presented in figures in the main article rather than as Supplementary Information. Indeed, most figures selected for inclusion in the main article present specific example cases (Figures 3-6), yet there is only one limited figure to illustrate overall performance via large scale analysis (Figure 2), so I feel an additional figure relating performance to relevant factors would be justified/appropriate.

Good point. We have moved the analysis on docking results by ligand size and resolution from supplemental information to Figure 2.

In Supp Fig 2 docking success is analysed as a function of map resolution. This is a very important analysis, but the way it is presented is misleading due to using the nominal resolution of the overall map rather than the local resolution of the map in the vicinity of the ligand. The result is that the dependency of

201 *docking success on map resolution is most likely highly underestimated. The authors conclude that*
*“docking success is resilient to changes in overall map resolution” (Page 3 line 98), which I feel is very*
*misleading, and quite possibly inaccurate. Ideally the authors should use local resolution for such*
*analyses - there are various tools available for calculating estimates of local resolution (e.g. ResMap).*

This is a fair criticism as there are several cases where the local resolution greatly differs from the
nominal resolution (Review Fig. 3). To more accurately reflect the environment where we are modeling,
we have switched to local resolution for our reporting. When looking at local resolution, we see a slight
correlation on local resolution for docking results. The distribution of resolutions for matched cases shifts
towards the higher side of the tested resolution range, the similar or better cases have a higher
distribution in the 4-5 Å range, and the worse case have higher representation at fringe resolution values.
The main text has been updated to include this analysis, and the methods have been updated to include
our local resolution calculations.

Review Fig. 3. Local resolution vs. nominal resolution for all 1053 cases in the dataset. Dashed line is
where the local resolution is 1.0 Å below the nominal resolution. The nominal resolution was used as the
local resolution for cases below the line.

*Still looking at Supp Fig 2, the decision to show density rather than frequency on the vertical axes is*
*understandable. However, the frequencies should be made available so that the subfigures can be more*
*fairly compared. I would suggest adding the numeric frequencies to the plots themselves, using the same*
*green/orange/blue colours. Or alternatively switch to showing frequencies on the vertical axes. It may also*
*be worth considering whether violin plots would be an informative representation.*

The difference in size across docking result groups is stark, so the density plots were used to show
differences in distributions while sharing the same axes labels. We agree that it can be misleading
though, especially since there are so few cases in the worse quality category. Now, we have included
stacked bar plots showing the percentage of docking results for bins of local resolution and number of
torsions.

*I would suggest also adding additional complementary visuals that will allow better analysis of the effect*
*of resolution and number of torsions on docking success. I would say that the quantity of interest is the*
*success rate - i.e. the proportion of successes versus failures - so this should be the focal quantity shown*
*on the vertical axis. If appropriate, differences in sample size for different resolutions/torsions can be*
*accounted for by adding confidence intervals based on standard error.*

As mentioned in the previous comment, we include stacked percentage bar plots by docking result across
resolution and torsion bins.

*It seems that there are very few cases below 5 Å resolution, and of those there seem to be more cases*
*where EMERALD fails than cases where it correctly docks the ligand. Consequently, the statement in the*
*Abstract that “this method is robust at predicting ligands in maps as low as 6Å resolution” is not supported*
*by the results presented. I note that the novel example presented in Figure 6 fits linoleic acid into a*
*cryoEM map with nominal resolution 2.85 Å. This issue of resolution is further exacerbated by confusing*
*the overall nominal resolution of a map, and map resolution in the context of ligand docking (i.e. local*
*resolution). Hopefully rethinking the presentation of the results (as per my comments above) will facilitate*
*the ability to make more precise statements that are backed up by appropriate levels of supporting*
*evidence.*

Thanks to your previous suggestions of considering local instead of nominal resolution, we now have
more cases lower than 5 Å resolution, going from 15 to 91. Additionally, the stacked bar plot in Fig. 2b
now shows that there are several successes in the lower resolution ranges. This strengthens the claims
we previously made that we can model in maps as low as 6 Å.

*The referencing of other works in the Introduction could be improved. Some of the references for tools*
*originally developed for macromolecular crystallography (MX) (references 10-14) are relevant but quite*
*old (2001-2007). Some of these programs have in more recent years been adapted for cryoEM - this*
*should at least be acknowledged, citing the corresponding more recent papers (e.g. Casanal, Lohcamp &*
*Emsley, 2019; Chojnowski, Sobolev, Heuser & Lamzin, 2021; etc.), irrespective of whether the authors*
*feel that those tools are unsuited to application at more moderate cryoEM resolutions. Note that the*
*authors’ use of nominal overall rather than local resolution makes such judgements about the suitability of*
*different tools at different resolutions ill-defined. Also, since there are no direct comparative performance*
*analyses of different tools in the present manuscript, nor any prior such analyses referenced, I would be*
*very careful about pigeonholing particular tools as being suitable only at certain resolutions without citing*
*due evidence, for fear of unintentionally causing offence via slander! (Did the authors try using other tools*
*to dock the linoleic acid shown in Figure 6?)*

While the referenced methods have been adapted for cryoEM, they have been adapted for protein
modeling and no specific changes for small molecules have been added to our knowledge. Regardless,
we have included this point in the introduction and added the proper citations for the cryoEM updated
versions. The resolution limits claimed in the manuscript are based on the poorer performance that the
methods cite at lower resolutions in their own manuscripts, and some methods do not show modeling
results at resolutions common for cryoEM. However, since that doesn't prove that they won't work at
lower resolution and because of the differences in resolution of MX and cryoEM, we have changed the
language around criticizing the performance of the cited methods at resolutions below 3 Å, saying they
are unproven rather than unusable.

*I also struggle with the comment: “But as resolution decreases below 3Å, the topological features these*
*methods rely on become less defined, and their accuracy in modeling ligands within 1Å RMSD of a*
*reference ligand falls below 20% [13, 14]”. My main problem with this is that the concept of resolution is*
*completely different in MX (resolution of the highest miller indices selected for use in refinement) and in*
*cryoEM (based on half-map FSC criteria), to the point where the numerical values cannot be directly*
*compared. References 13-14 are from 2006-2007 and refer to application in MX, and thus the quoted*
*conclusions do not transfer directly to cryoEM. Note that the fact that cryoEM “data” include phase*

*information has a positive effect on the quality of the maps relative to MX, in contrast with MX where*
*phase information is lacking and the maps are conditional on the current state of the model (and there are*
*also other relevant differences between MX and cryoEM maps). Indeed, for a given nominal resolution,*
*MX maps tend to be much worse than cryoEM maps overall, meaning that it might be argued that ligand*
*fitting is generally a harder challenge in MX than in cryoEM.*

We understand the concern of comparing resolutions and difficulties in modeling with MX and cryoEM. As
the local resolution analysis shows, there are several cases in our dataset that are at 4 Å resolution or
worse, which regardless of discrepancies between MX and cryoEM is well below the quality of map
shown to be tested on the methods cited. So when the cited manuscripts have poor accuracy below 2.5-
3.0 Å resolution, it is reasonable to expect them to perform poorly at low resolution cryoEM as well. Also,
a lack of shape features to guide modeling is relevant regardless of a MX or EM map, as for both types of
maps lower resolution results in less defined maps.

Additionally, the updates of the cited methods to consider cryoEM do not state any changes to ligand
protocol, so if there are such large discrepancies between MX and cryoEM maps, then there is an even
bigger need for cryoEM tailored small molecule modeling.

*In the context of discussing other tools in the Introduction, it might be worth also acknowledging that the*
*various tools have been developed with different intended use cases in mind - in particular, the tradeoff*
*between computational performance and docking quality. It is quoted that the present tool takes 30-*
*120mins to run (Page 2 line 75), whereas some other available ligand fitting tools might take just a few*
*seconds to run. There is a difference between a quick attempt at identifying reasonable initial conformer*
*(that would require further manual intervention and/or model refinement), versus a more computationally*
*intensive fully automated solution to docking (including refinement of the structural environment). It seems*
*that EMERALD is intended to perform the latter.*

Yes, this is a good point, and EMERALD is indeed intended for the latter. The text has been updated to
make more explicit the intended use of EMERALD to produce an unbiased, deposition-ready model with
little user input in the discussion.

*That being said, from the “Docking protocol and analysis” section in the Supplementary Material it is not*
*completely clear whether the protein model in the vicinity of the ligand is also refined in order to optimise*
*the protein-ligand stereochemical interactions, or whether the ligand is simply docked without modifying*
*the protein model - this should be clarified, adding more methodological/implementation details.*

The methods have been updated to make it clearer that the entire pocket is optimized along with the
ligand conformation.

*One weakness is that I see no mention of half-maps. Is there a way that half maps can be used to*
*complement the approach, for validation / assessment of confidence in the docked ligand model?*

Half-maps were not originally considered because only 246 maps in the dataset had associated half-
maps. With the EMDb now requiring half-maps with deposition, you're right that half-maps should be
considered and shown how they could be utilized with EMERALD results. One thought is that half-maps
could be used to identify overfitting, as previously shown (DiMaio et al. *Protein Science* 2013). The
deposited models and docked models were refined into a training half-map, and then the density cross
correlation was determined using the other validating half-map. Maps with large gaps in density
correlation could be flagged for potential overfitting. For many instances with large gaps, the difference in

density agreement with the training and validate map is because the map in one map is much weaker
(Review Fig. 4a-b). Although, it is possible to find instances of overfitting (Review Fig. 4c-d).

Review Fig. 4. Half map analysis for ligand overfitting. (a,b) Differences in resolution between the training
map (a, blue) and the validation map (b, gray) can lead to incorrect evaluation that a ligand is overfit to
the map. (c,d) Despite this there was one case, RI5 in 6UD3, where the EMERALD model (blue) appears
to be overfit to the training halfmap (c) than the deposited model (white), while the deposited model
correlations better with the validation map (d).

Another way to use half-maps is to see if our converged, similar-or-better docked models fit in an
independent half-map better than the deposited model. We found that large deltas in density correlation
between the deposited and docked model on a validation half-map could parse out ligands with improved
models (example in Fig. 4 c-d), or could indicate false positive improvements (Suppl. Fig 3). The text has
been updated and a supplemental figure has been added to include the halfmap analysis, and Fig. 4c-d
has been updated to include a case from the half map analysis.

*In the cases of ambiguous maps (e.g. Figure 5), did the authors consider using complementary maps in*
*order to attempt to further critique the docked ligand models? For example, locally sharpened maps (e.g.*
*LocScale) and difference maps (e.g. Servalcat); there may be other similar tools for model/map analysis.*

We did not consider other map processing techniques. While Suppl. Fig. 4 highlights a case where
alternate map processing can be useful in rescuing a failure, limited empirical testing shows that: a) this is
largely map-dependent, and b) the algorithm is not extremely sensitive to map processing. For this
reason, we opted to use the deposited maps to illustrate our approach works reasonably well over a
range of map processing techniques.

*On pages 4 and 5, in reference to Figures 3 and 4, it is not sufficiently clear whether the references to*
*finding “alternate conformations” implies that the deposited models are wrong and that the EMERALD-*
*docked ligands are correct, or whether it is implied that both the deposited and EMERALD-docked ligands*
*are correct / plausible conformations. The authors’ intended interpretation should be clarified, just to make*

*this completely explicit for the reader. In this context I will point out the comment in the Discussion on*
*Page 7 line 242: “in some cases correcting errors present in the deposited model”.*

Given the low resolution where we are working, it is difficult for us to claim that a certain model is incorrect
unless there is support from a higher resolution structure. So, we do mean that our models offer a
plausible conformation to explain the density, without claiming that the deposited model is wrong. All
instances of calling our model correct or the deposited model incorrect have been changed to avoid
confusion. Between comparisons to crystal structures, the above half map analysis, and EMERALD
convergence, the level of confidence and support for particular models are known throughout the paper.

*But for now, I will assume that the authors intend for the reader to understand that the identification of*
*alternate conformations does not necessarily mean that the deposited conformations are wrong, as*
*multiple conformations may actually be present. However, if proposing a multi-conformer binding site*
*model, then appropriate validation should be performed in order to verify that it is better than a single-*
*conformer model. When publishing works such as this, there is a certain responsibility to demonstrate*
*best practice so that potential users can follow suit in practical application; in that sense, the current*
*status of the presented examples (Figures 3 and 4) is a half-finished solution. In order to further verify and*
*model such multi-conformer cases, I would suggest overlaying the conformers, modelling them together*
*by assigning different alt conf IDs, and performing refinement of the binding site (including refinement of*
*occupancies). Only then, by comparing the RSCC values and difference maps, would it be possible to*
*claim that the modelling of the binding site is improved by modelling the ligand in multiple alternate*
*conformations.*

This is a tricky point to address. For the majority of cases with differences, even though both models
match the data equally well, EMERALD prefers one particular conformation, likely due to the underlying
forcefield. However, that does not distinguish between a mistake in the original structure or the presence
of multiple conformations. So we do not try to make that distinction. While we could try multi-
conformation refinement, it makes the comparison challenging as allowing additional parameters in
refinement will almost certainly lead to improved RSCC/difference maps.

In addition to our analysis of crystal structures, what we have done to address this in our revision is to use
half maps for additional validation in cases where they were available (see above). This suggests that 7
out of 63 have strong evidence for correcting a mistake in the original structure; the remainder still remain
ambiguous.

*Specific comments by page/line:*

*Page 1 line 7 - Affiliations 1 and 2 appear to be the same.*

Updated in the manuscript so affiliation 2 is “Institute for Protein Design”

*Page 1 line 22 - It seems strange to refer to “This tool” in the Abstract, without mentioning the name of the*
*tool (EMERALD).*

Good catch. The abstract mentions EMERALD now.

*Page 2 line 47 - The comment “However, both require user input in either selecting models” reads like a*
*criticism of those approaches... so is it implied that EMERALD doesn’t require the user to select a*

*model?! The Discussion on Page 7 line 255 implies otherwise: “our method requires the modeler to know*
*the identification and approximate binding location of the ligand”.*

The intention behind the introduction sentence on page 2 line 47 is that the methods cited lack full
automation. In steps within the pipeline, they require human analysis and intervention on the way to
determining a final model. For EMERALD, while the identity of the ligand and approximate binding pocket
is needed, the protocol will automatically produce a final ligand model with no input or analysis. The text
has been updated to indicate the human intervention throughout the pipeline on page 2 line 47 to make
the intentions clearer.

*Page 2 line 52 - Inconsistent use of “force field” vs “forcefield”.*

All instances are changed to “force field” now.

*Page 2 line 57 (and elsewhere) - I would suggest using the word “map” rather than “density” throughout*
*the manuscript, given that maps output from cryoEM reconstructions correspond to electrostatic potential,*
*not (electron) density.*

It is common in EM to call an EM map a (cryoEM) density map, despite that it is not technically electron
density. Because of this we feel it is appropriate to leave instances of “density”.

*Page 2 line 75 - It is good to give an indication of computation time (30-120 minutes). However, it would*
*be interesting to know how that time is partitioned into the different steps? Is there one step that*
*dominates the computation time? Is overall time essentially determined by ligand size, or is the protocol*
*sensitive to decisions/parameters such as the number of conformers to trial?*

The time scales based on size of the ligand. The initial population for the genetic algorithm is
oversampled, so that is the step with the most variation as it takes longer to randomize ligands with more
degrees of freedom and will take longer as the population size is increased. Population sizes are equal
for all runs in the manuscript, so the variation in runtime comes from the difference in ligand size). The
manuscript has been updated to mention the time intensive step in the “Docking protocol and analysis”
portion of the methods.

*Page 3 line 102 - In this particular context, “low resolution” should not be hyphenated. It should either be*
*“low resolution of the maps” or “low-resolution maps”.*

Good eye. The hyphenation/lack of hyphenation of “low resolution” has been corrected here and other
places in the manuscript.

*Page 4 line 127 - I’m confused about the switch from “1Å” to “1.0Å”? Also note that technically there*
*should be a space between the numeric value and the unit (e.g. “1 Å”)... I won’t list all instances of this.*

Manuscript is updated to consistently use 1 Å. Spaces have been added between the number and unit
throughout the text.

*Page 4 line 136 - “are” should be “our”.*

Mistake has been corrected

*Page 4 - inconsistency writing "Figure" or "Fig." when referencing figures.*

Updated in the text.

*Page 4 line 142/143 - I don't think it's necessary to refer to Fig 3d twice here.*

Removed the second reference in the text.

*Page 7 line 257 - I don't think it's appropriate to say "our method could expand to recognize potential*
*unmodeled ligand blobs and quickly assess possible ligands to determine identity", given that it is not*
*clear how the method could be straightforwardly expanded to provide those functionalities. Rather, it is*
*sufficient to simply state that such functionalities would be desirable.*

We updated "our method could expand to recognize potential unmodeled ligand blobs and quickly assess
possible ligands to determine identity" to "expanding our method to recognize potential unmodeled ligand
blobs and quickly assess possible ligands to determine identity would be beneficial."

*Figure 1 - The caption would benefit from some more verbosity. In particular, in subfigure (a) the red*
*cross and black circle lack explanation, and the blue arrows in subfigure (c) that represent the genetic*
*algorithm are too cryptic.*

More detailed explanations were added to the figure legend.

*Figure 2 - "4% were 1Å RMSD of" should be "4% were more than 1Å RMSD from".*

Corrected in the manuscript.

*Figures 3-6 - Unnecessary duplication between the figure captions and the main text in the Results*
*section. The figure captions should be an objective description of the contents of the subfigures, whereas*
*the main text should provide a more detailed narrative - the overlap between these should be minimised*
*where possible. Also, I do not think it is necessary to provide EMDB and PDB codes and citations in both*
*the main text and figure captions - I would lean towards providing these only in the figure captions in*
*order to improve readability of the main text.*

The EMDB and PDB codes have been eliminated in the main text and only appear in the figure legends.
The figure legends have also been updated to only state the cases shown in the figure, leaving the
analysis to the main text.

*Figures 3-6 and Supplementary Figures 3-4 - for all EMDB entries used as examples the nominal*
*resolutions should be stated. This is relevant, in order to give the reader an indication of relative map*
*interpretation difficulty. Providing a measure of local resolution would be even better, for the reasons*
*stated above.*

The calculated local resolution for every case shown has been added to the figure legends.

*Given that a number of improved (or alternative) ligand poses have been identified during the course of*
*the present work, have the authors reached out to the original depositors of the PDB models in the*
*interest of improving the quality of the models in the database and derived biological interpretations?*

We have not reached out to any authors. We would be willing to do so and work with them.

Reviewers' Comments:

Reviewer #1:

Remarks to the Author:

The authors have made a serious effort to address the issues raised in the previous round of reviews. I do not have strong opinions for its publication in Nature Communications one way or the other.

Reviewer #2:

Remarks to the Author:

I feel the authors have addressed well the points of all reviewers and the manuscript is much improved as a result. I recommend publication.

Reviewer #3:

Remarks to the Author:

The authors have conscientiously addressed the comments and revised the manuscript accordingly. Presenting local rather than nominal resolution provides a more informative context.

Given that the ability to predict ligands in maps as low as 6 Å resolution is one of the main claims, it is a shame that the lowest local resolution example presented in the figures is 4.42 Å, thus making this claim somewhat tentative. I would suggest considering whether there are any convincing lower local resolution examples that could be included in Figure 3; and if a good example cannot be found then I would suggest tempering the claim of useful practical application at 6 Å down to a resolution that has been clearly evidenced.

Figure 2b does demonstrate dependence on local resolution, clarifying that the implementation can produce ligands with similar conformations to those in PDB models in a substantial proportion of cases (including at low resolution). However, as Reviewer #1 points out, the ability to match/reproduce a conformer in the deposited PDB model does not necessarily mean that the conformer is correct/optimal, and, as Reviewer #2 points out, it is always important to critique stereochemical details of the protein-ligand complex. With that in mind, I would be cautious of claiming that the tool "automatically produces deposition-ready small molecule models" unless also providing extensive and rigorous validation as part of the output.

Continuing to think about validation, it is good that half maps have been used to investigate overfitting in the dataset. I cannot help but notice that the authors' response to my comment on this topic is much more verbose than what has been added to the main text. Indeed, Review Fig 4 provides a good warning example. More generally, it might be worth considering adding an instructive narrative of the authors' recommendations RE post-EMERALD ligand validation (use of half-maps and other considerations) to the Discussion.

At the beginning of the Discussion (Line 244), I am confused about the exact meaning of the phrase "without bias" that has been added during revision... without bias from what? There are various types of bias, and there is nothing to support/clarify this statement - bias is not mentioned elsewhere in the manuscript. If the input map and/or starting model is biased/incorrect in any way then this will undoubtedly affect ligand modelling efforts.

I am glad that the ligand models that were produced as part of the analysis have been made available for download and further inspection. Overall, I am satisfied that the article has been substantially improved as a consequence of addressing the reviewers' comments. As a useful reference describing the EMERALD protocol and setting expectations of performance, I can recommend it for publication

following consideration of the above comments.

Robert A. Nicholls

Reviewer #1 (Remarks to the Author):

The authors have made a serious effort to address the issues raised in the previous round of reviews. I do not have strong opinions for its publication in Nature Communications one way or the other.

Reviewer #2 (Remarks to the Author):

I feel the authors have addressed well the points of all reviewers and the manuscript is much improved as a result. I recommend publication.

Reviewer #3 (Remarks to the Author):

The authors have conscientiously addressed the comments and revised the manuscript accordingly. Presenting local rather than nominal resolution provides a more informative context.

Given that the ability to predict ligands in maps as low as 6 Å resolution is one of the main claims, it is a shame that the lowest local resolution example presented in the figures is 4.42 Å, thus making this claim somewhat tentative. I would suggest considering whether there are any convincing lower local resolution examples that could be included in Figure 3; and if a good example cannot be found then I would suggest tempering the claim of useful practical application at 6 Å down to a resolution that has been clearly evidenced.

Unfortunately, there are no cases where a crystal model confirms the EMERALD docked model at a resolution around 6 Å. There are a handful of confidently docked EMERALD models that match the deposited EM model at local resolutions down to 6 Å, but like you mention in the next comment, matching the deposited model does not necessarily mean the model is optimal. We have tempered the resolution claims to 4.5 Å with readers being able to see that we can match the deposited model as low as 6 Å by observing Fig. 2b.

Figure 2b does demonstrate dependence on local resolution, clarifying that the implementation can produce ligands with similar conformations to those in PDB models in a substantial proportion of cases (including at low resolution). However, as Reviewer #1 points out, the ability to match/reproduce a conformer in the deposited PDB model does not necessarily mean that the conformer is correct/optimal, and, as Reviewer #2 points out, it is always important to critique stereochemical details of the protein-ligand complex. With that in mind, I would be cautious of claiming that the tool "automatically produces deposition-ready small molecule models" unless also providing extensive and rigorous validation as part of the output.

We believe that because several of our small molecule models were confirmed by high-resolution crystal models and/or match the deposited EM model that EMERALD is able to produce models at the quality for deposition. However, as pointed out, the deposited models may not be correct and there are still cases where

EMERALD does not produce a model as good as the deposited model. We have changed the wording to say EMERALD is capable of a deposition-ready mode, rather than implying that it can always be used to produce a deposition-ready model without inspection.

Continuing to think about validation, it is good that half maps have been used to investigate overfitting in the dataset. I cannot help but notice that the authors' response to my comment on this topic is much more verbose than what has been added to the main text. Indeed, Review Fig 4 provides a good warning example. More generally, it might be worth considering adding an instructive narrative of the authors' recommendations RE post-EMERALD ligand validation (use of half-maps and other considerations) to the Discussion.

We agree it would be worthwhile to include recommendations since half-maps are available to those performing modeling. However, we ultimately performed our half-map analysis on a small subset of structures which makes it hard to provide confident recommendations in the discussion.

At the beginning of the Discussion (Line 244), I am confused about the exact meaning of the phrase "without bias" that has been added during revision... without bias from what? There are various types of bias, and there is nothing to support/clarify this statement - bias is not mentioned elsewhere in the manuscript. If the input map and/or starting model is biased/incorrect in any way then this will undoubtedly affect ligand modelling efforts.

We were referring to human bias from the modeler while building the small molecule. As you point out this was vague and unclear so the manuscript has been updated to make this point clear.

I am glad that the ligand models that were produced as part of the analysis have been made available for download and further inspection. Overall, I am satisfied that the article has been substantially improved as a consequence of addressing the reviewers' comments. As a useful reference describing the EMERALD protocol and setting expectations of performance, I can recommend it for publication following consideration of the above comments.

Robert A. Nicholls